# DOCUMENTARY EVIDENCE OF DROUGHTS IN SWEDEN BETWEEN THE MIDDLE AGES AND c1800

**Lotta Leijonhufvud and Dag Retsö**

**Abstract**

This article explores documentary evidence of droughts in Sweden in the pre-instrumental period (1400-1800). The database has been developed using contemporary sources such as private and official correspondence letters, diaries, almanac notes, manorial accounts, and weather data compilations. The primary purpose is to utilize hitherto unused documentary data as an input for an index that can be useful for comparisons on a larger European scale.

The survey shows that eight sub-periods can be considered as particularly struck by summer droughts with concomitant harvest failures and great social impacts in Sweden. That is the case with 1634-1639, 1652-1657, 1665-1670, 1677-1684, 1746-1750, 1757-1767, 1771-1776 and 1780-1783. Within these sub-periods, 1652 and 1657 stand out as particularly troublesome years. A number of data for dry summers are also found for the middle decades of the 15[th] century, the first decade of the 1500s and the 1550s.

*Introduction*

The purpose of this paper is to present documentary evidence of drought in Sweden for the period 1400 to 1800. We also try to present a link between instrumental data from precipitation and temperature to our drought index. Is it possible to distinguish periods of drought in Sweden through documentary sources from the 15[th] till the 18[th] century?

Stretching from 55° N to 69° N Sweden is characterized by arctic climate in the extreme north and temperate climate in the south. Located between the Baltic Sea and the Scandian mountain range wet weather from the Atlantic affects the western part of Sweden, while the eastern part is protected both by the Scandian mountain range and highland in the south, rendering average precipitation in the eastern part between 300 and 700 mm a year, compared to western part which ranges between 800 and 1200 mm a year. The length of the winter and the length of the growing period, which varies in a southern-northerly direction, have the most distinct effect on agricultural production and society in general. Still, the early modern history of Sweden gives evidence of repeated periods of severe droughts.

In general, drought at the latitude of Sweden is caused by deficient precipitation and only occasionally by excessive temperature and evapotranspiration. Sometimes several meteorological and hydrological factors do combine to produce severe drought with serious socioeconomic consequences. For example, apart from deficiency in precipitation (meteorological drought) seasonal lack of streaming water can also be the result of late spring or low summer temperatures in the Scandian mountain range when snow fail to melt at a normal pace resulting in insufficient discharge into the rivers which produces streamflow (hydrological) droughts and/or low flows (Hisdal and Tallaksen 2000). Insufficient spring floods also partly lies behind failed harvests of hay grown in wet meadows and in historical times concomitant raised cattle mortality. Conversely, low water levels in streams due to dry autumn/summer weather facilitates quick freezing in the early winter and implies further obstacles to running watermills. Therefore, in the long run droughts do affect agriculture but strike more directly at industrial activities depending on water power. Socioeconomically this

has had serious consequences for Sweden, to a large degree dependent on mining and exports of iron and copper especially from the 17<sup>th</sup> century onwards.

*Sources*

The indices used in this paper have been constructed from a database launched by prof. emer. Johan Söderberg, Department of Economic History, Stockholm university, where both authors of this paper have contributed too through adding weather information by excerpting original data from letters, chronicles, newspapers etc. The database consists of a wide variety of documentary sources: diaries, official letters, chronicles, as well as published articles in papers from the Royal Swedish Academy of Sciences and early newspapers. This database will be available to the public through the Bolin Centre later this year (only in Swedish though.) The database has some 20,000 entries from 1500 to 1870.

A typical statement of a severe drought is found in the diary of the parish priest Petrus Magnii Gyllenius, who also made summarized descriptions of entire years, in the province of Värmland. For the year 1652 he writes (our translation): "In the beginning of May it rained a little. Then there was a great drought, this year was called the great drought year. No rain fell, neither in Sweden or Finland, between early May and late September, with the exception of 25 June when some thunder rain fell over Letstigen in [the province of] Närke, as on 30 June when it rained a little in Karlstad. In Sweden there was a quite great harvest failure this year for grain due to the severe drought and heat. The drought destroyed the grain in many places, so that nothing was saved of the spring seed, and there were dear times. At the same time there was little hay […] Forest fires caused great damages in Sweden and Finland. Bridges and hay barns burned" (Hausen 1880: 198-201).

*Instrumental measurements*

In this study we have used homogenized historical instrumental data from Stockholm observatory. The temperature record begins in 1756 and precipitation data in 1786. The first thing we wanted to do was to examine if there was any relationship between precipitation/drought and temperature since precipitation data before 1859 seem more unreliable than after that year: the data are not represented with decimals and correlation coefficients between precipitation and temperature become non-existent. Precipitation data before 1893 also exhibit severe under-catch problems (Moberg et al, 2003: 1501). Moberg et al adjust precipitation data with different factors, which we have not done, since our focus is the drought index and the kind of factor increasing adjustments done there will have no effect on correlation coefficients.

*Method*

In this article, the annual indices from documentary data have been based on the stated intensity of the drought event and its spatial extension. It has only been possible to construct reliable indices for the 16<sup>th</sup>, 17<sup>th</sup> and 18<sup>th</sup> centuries since no continuous time series can be reconstructed before that due to insufficient amounts of documentary records. Nevertheless, an overview of documentary data from the 15<sup>th</sup> century will be given.

For some years, the documentary data are too contradictory to enable any definite conclusions. In some cases, it derives from regional variations. One example is from 1554, when there was "severe drought" in the province of Uppland and at the same time good harvest in the Kronoberg province further to the south (Forssell 1884, bil A: 161). But even when data are relatively plentiful, they can be contradictory. One such example is the year 1733. Some

data from that year speak of an "unusual" drought in the provinces of Västergötland in the west, and Hälsingland and Dalarna further to the north in May (Broman 1911-1949: Olofsson and Liedgren, 1974: 261). In a period of 18 weeks between early March and the end of June only three short showers of rain are said to have fallen in Västergötland, a province with typical humid weather conditions, and the water level of Lake Vänern was quite low (Bergstrand 1934: 196; Wallén 1910: 13). At the same time the harvests were good in general in Sweden and there are no reports of harvest failures (Utterström 1957: 429). In Västergötland itself the harvest was even said to have been plentiful (Olander 1951: 119). The explanation for this discrepancy may be different timings of sowing of different crops, where e.g. early-maturing crops like barley and wheat (the latter of those was cultivated in Sweden only to a small degree before the 19[th] century) (Söderberg and Myrdal 2002) suffered most and crops with a long growing season, like rye and buckwheat, could survive. In no case there are evidence of droughts covering the entire growing season, which means that no generalized nutritional catastrophe has been registered. A mitigating factor was that periodically local demand for foodstuffs was reduced through the absence abroad of a large part of the male population in the numerous wars Sweden fought in Europe between 1563 and 1718.

The most important part of the present analysis is the construction of an index. The construction was made comprehensively so that notices on drought or precipitation were evaluated within the context of the database.

As can been seen in Figure 1, there are many more notices which we have labelled "dry", especially in the 18[th] century, than there are notices on "wet" conditions. The word "rain" occurs 3,361 times in the database (of a total of 20,896 entries), while the word "sun" only occurs 1,224 times. However, varieties of "heat", "dry", "warm" occur 1,726 times compared to the two words describing "wet" in Swedish, which only occur 292 times. Many notices regarding rain are of the kind "A beautiful rain fell"; suggesting that rain was welcome. Generally, wet conditions are defining for agriculture in Scandinavia, but many fields are located such that they have a natural drainage (Leijonhufvud 2001: 130). These findings suggest that although notices of rain are more frequent than notices describing fine weather, consequences of "fine" weather were more troublesome. Figure 1 depicts the drought/precipitation index that has been constructed. Positive signs indicate descriptions of droughts that have caused problems or concern and negative values indicate years when precipitation have been the cause for such impressions. Superimposed is a 9-year quasi-Gaussian smoothing filter.

Fig. 1. Drought/precipitation index 1500–1816

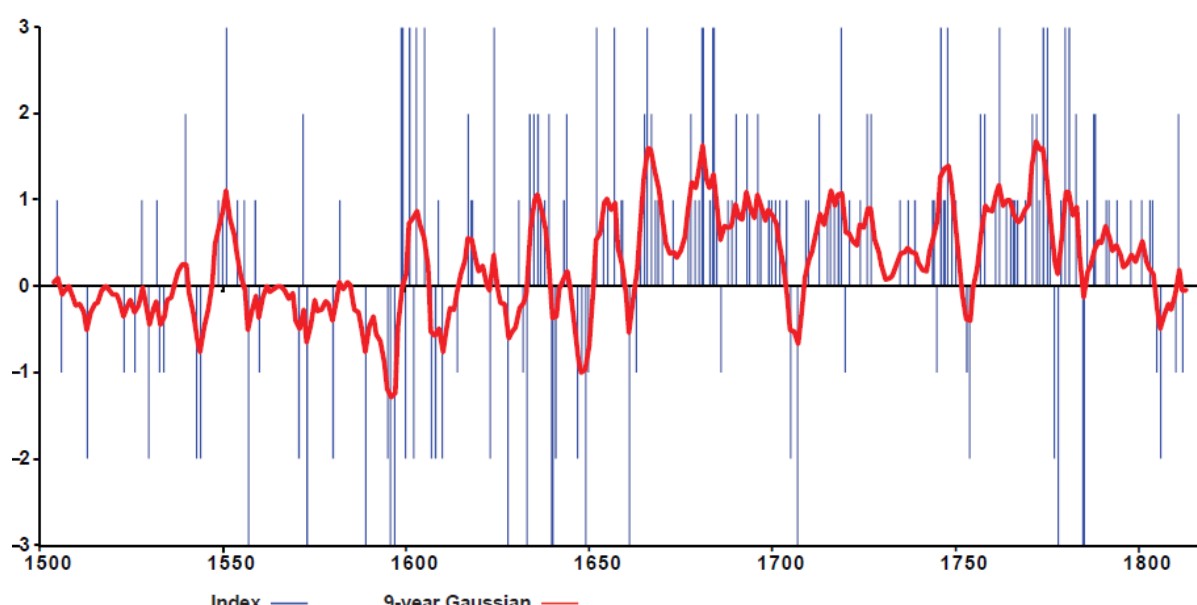

Note: We wish to thank associate professor Fredrik Charpentier Ljungqvist at the Stockholm University for his help with the graphs.

Correlation between proper instrumental data of temperature and precipitation, from the same observational site of Observatorielunden in central Stockholm showed, rather surprisingly, a slightly negative correlation between summer (JJA) temperatures and precipitation of -0.35. This result is similar to Moberg et al 2003, Table VI, which is higher, probably because of a slightly different period (1873-2000).

Since there are no reliable precipitation/drought data before 1860, we have tested the index against the Stockholm temperature series from 1756. Correlation between the index and average monthly temperature for the period 1756-1816 turned out significantly for the months May, June and July (Table 1) with the highest correlation received using MJJ-temperatures. However, since correlation for instrumental data between precipitation and temperature in May was very weak (non-existing), we argue that the standard season of summer months (June, July, August) will be more adequate in our exploration of droughts. Correlation between the index and average monthly temperature for the period 1756-1816 turned out significantly for the months May, June and July (Table 1).

Table 1: Correlation between average monthly temperatures against the drought index 1756-1816 and precipitation of Stockholm for the period 1859-2011 (daily observations calculated to monthly values).

| | April | May | June | July | Aug | MJJ | JJA | C-Scan |
|---|---|---|---|---|---|---|---|---|
| Index | 0.26 | 0.30 | 0.52 | 0.38 | 0.13 | 0.51 | 0.47 | 0.08 |
| Precip | -0.15 | -0.24 | -0.30 | -0.38 | -0.33 | -0.25 | -0.35 | 0.20 (Jan) |

Precipitation data were downloaded from https://www.smhi.se/data/meteorologi/ladda-ner-meteorologiska-observationer/#param=precipitation24HourSum,stations=all,stationid=98210 on 22nd January 2020

*Documentary data on droughts for the 15th and 16th centuries*

For the 15th and 16th centuries documentary data are scarce, uneven and spread out in a number of different source categories. Indeed, there is a number of evidences for harvest failures, although the reasons are rarely stated (Retsö 2015). It is possible that prices of certain

other goods contain climatological information, in particular wax and honey, both highly
dependent on weather conditions in the summer. During the last three decades of the 16[th]
century, production of bee wax was much reduced in Sweden, probably due to the transition
to a cooler and wetter climate which was damaging to the bees (Husberg, 1994). Further
archival research is needed to expand wax price series needed for climatic research. Grain
prices seem to be more associated with temperature than drought variability (Charpentier
Ljungqvist, 2021, in print).
Data on agricultural activities in the province of Ostrogothia are found for a few
years in the first and last decades of the 15[th] century. Harvesting dates for 1402, 1407 and
1410 suggest close to normal summer temperature and precipitation (Lundén, 1958: 141, 161,
133; Retsö, "Normality and anomaly", in preparation) while available data on dates for
sowing of barley and other grains and fodder for swine indicate somewhat late or cold spring
in 1491, and early or warm spring in 1489, 1490 and 1492 (Alvered, 1999: 104, 145, 192,
245).
Food crises are frequently mentioned in the 15[th] century, in particular the four decades
between 1430 and early 1470s. It is assumed here that the mentioning of a food crisis in a
particular year reflects a harvest failure the preceding year. As for the 1430s, we know that a
period of crisis years began in 1435 and although we have no Swedish evidence of dearth for
the first years of the decade, it can be noted that Danish and German sources mention hard
times and high corn prices in 1433 that could be connected to cold springs (see Camenisch et
al., 2016: 2110). It is also conspicuous that a major peasant uprising occurred in Sweden in
1434 and it can be suspected that it had something to do with a food crisis in combination
with unusually high taxes. In the spring of 1437, there was a lack of food grains in Finland
and famine and dearth in Sweden are mentioned in early 1438 (Hausen, 1921 no. 2220;
Tunberg, 1937: 214). The monetary valuations of the barley tithes in Funbo parish in Uppland
in 1438 and 1439 more than doubled compared to the preceding years (Andræ, 1965). These
years are well-known in continental Europe as a time of food crises with concomitant social
and economic impacts. The harvests of 1437 and 1438 were the worst in England during the
15[th] century, and the price of grain rose to an exceptionally high level in 1439. The famines of
the mid-1400s occurred in a context of repeated plague epidemics also hitting Sweden
(Myrdal, 2003: 249). They also fall within a subperiod of colder summers related to a Spörer
minimum of solar activity within a longer period (1400-1550) of slightly warmer summers as
compared to the 20[th] century, at least in northern Fennoscandia, according to tree-ring data;
the eruption of Mount Fuji in 1435/1436 in Japan may have contributed to cold winters and
late and cool summers in north-western Europe during these years (Moberg et al., 2006: 24,
26ff; Campbell, 2009: 30; Camenisch et al., 2016: 2110).
The 1440s were also troubled by harvest failures. In 1442 the rye and hops harvest
failed in Finland (Hausen, 1921 nos. 2512 and 2517; Bunge and Hildebrand, 1889 no. 955.
See also Hausen, 1921 nos. 2521, 2528, 2529, 2535) and just a few years later the Vadstena
abbey was forced to sell some of its valuable chalices and shrines in order to buy food, due to
the harvest failures in 1445 and 1446 (RA = Riksarkivet (National Archives of Sweden),
Stockholm, Medieval codex A21 fol. 89r-v). From 1446 there is information on famine in
Sweden (Hadorph, 1674: 370ff) and 1448 was described as a year of dearth in Stockholm due
to a dry spring and much rain from late May onwards (Klemming, 1866: 255).
The Vadstena annals describe the years 1454-1457 as struck by famine, which in the
first of these years was combined with an outbreak of plague (Gejrot, 1996: 286f, 292f;
Styffe, 1870: 85. See also Christensen, 1895: 297 n. 2; Fant, 1818: 173, 175; Codex dipl. lub.
1:9, no. 328; Ropp, 1883 nos. 516, 520) and in 1470 there was famine in Finland (Hausen,
1924 no 3142). This, as well as the harvest failure of 1460, may have had something to do
with a volcanic eruption in the Pacific in 1453, marking the onset of a 15-year cool period
(Esper et al., 2017).
Also the early 1470s display evidence of a period of hot and dry weather, apparently an
all-European phenomenon (Camenisch et al., 2020). In August 1474 the council of the
Swedish realm issued a statute regulating the use of watermills due to repeated droughts, i e
presumably causing lack of water (Hadorph 1676 no. 9). Furthermore, food crisis is indicated
in a letter from Åbo (Turku), Finland, from May 1471 (Hausen 1890 no. 625), in Sweden
nominal grain prices display an unprecedented peak in the early 1470s, (Franzén and
Söderberg 2006) and the Danish Roskilde annals speak of a "severely hot and burning
summer" in Denmark in 1473 (Rørdam 1873).
Summarizing, the years in the 15th century with harvest failures and/or unusually early
onsets of the growing season are the following: 1402, 1405, 1436-1437, 1439, 1442, 1445-
1446, 1448, 1453-1456, 1460, 1469-1470, 1473-1474, 1489, 1490 and 1492.
From the first decade of the 16th century there are a number of reports of harvest failures
and famine. In Västergötland, Småland and the Stockholm area they speak of unsown fields,
starving peasantry forced to eat bark, and expensive corn that point to a harvest failure in 1503
(RA Sturearkivet nos 255, 637; Styffe 1875 no 232). Shortage and poverty among the peasants
is reported for the following year (Wegener 1866-1870: 319-20). In southwestern Finland the
harvest of 1507 had been consumed already in July 1508 and the peasantry suffered famine and
"ate more bark than ever" (Hausen 1930 nos. 5324, 5329). Similar reports are found for the
same year from mid-Sweden and the Stockholm area (RA Sturearkivet nos 573, 597). 1508
seems to have been even worse. Again, prices on rye were high in March 1509, but already by
harvest time in 1508 prices were rising in Finland and the misery was said to be the worst in
ten years; by the end of the year the country was ravaged by both great poverty and plague,
unabling the peasantry to pay their taxes (Sjödin 1937: 336; Hausen 1930 nos. 5341, 5347,
5354, 5368). The same was reported from Sweden; in March 1509 the peasants northeast of
Stockholm starved and ate bark (Sjödin 1937: 322, 344, RA Sturearkivet no. 1053, Styffe 1875
no 229). Widespread poverty was also reported as a result of a bad harvest in 1509, already in
December in central Sweden, and in the spring and summer of 1510 (Sjödin 1937: 350; Styffe
1875 nos. 302, 304, RA Sturearkivet no. 1467).
In both Finland and south-eastern Sweden there was severe drought in late spring and
summer of 1551 (Almquist 1905: 115ff, 123ff, 212f, 430ff). Also, in the autumn there was a
severe drought in the Bergslagen mining area (Almquist 1905: 430ff, Johansson 1882: 159f).
In June 1559 the harvest of both rye and barley in Östergötland and southeastern Småland were
in danger already in its blooming time due to both night frost and drought (Almquist 1916: 190,
202, 651). The same was reported from Finland in September (Almquist 1916: 287). Apart from
1551 and 1559 there are also reports from other years of the 16th century but they are sporadic
and it is uncertain as to how extensive the droughts were. In 1599, there are evidence from
southeastern Småland of severe heat and forest fires (Edman 1985: 74; see also Utterström
1955: 29, Hallendorff 1902: 79) and the production of honey was reduced drastically (Husberg
1994: 275).
*Documentary data on droughts for the 17th and 18th centuries*
For the 17th and 18th centuries sources are far more abundant and continuous, among other
things thanks to a number of private diaries. Some periods stand out as particularly hit by
moderate to extreme drought. That is the case with 1634-1639, 1652-1657, 1665-1670, 1677-
1684, 1746-1750, 1757-1767, 1771-1776 and 1780-1783 (with two years of extreme drought
each) and 1634-1639 (with one year of extreme drought). Among these, 1652 and 1657 stand

out as particularly troublesome. Other single years seem to have been dry on an all-European scale, like 1540 (Wetter et al 2014). Although some of the dry periods recorded in Sweden coincide with similar drought episodes in other areas of Europe (see e g Brázdil et al., 2016), negative spatial correlations are to be expected between northern and southern Europe.

Eight periods stand out as particularly critical in terms of drought in the 17th and 18th centuries (for references for the particular years, see Table 4 below).

1) 1634-1639. There are reports of drought from the north as well as the south every year in this period. Weather conditions are characterized in the relatively detailed sources as generally dry with a typical pattern of dry and cold springs, hot and warm summers and rather wet autumn seasons. The result was disaster for the harvest of hay but rather good harvests of rye. The hardships could even have begun earlier than 1634; in June 1635 Gabriel Gustafsson Oxenstierna wrote to his brother that poverty was widespread in the whole country after "the last years [i.e. plural] of dearth" (Sondén 1890: 363).

2) 1652-1657. 1652 was called the Great Drought Year already in contemporary sources. Several reports from virtually all regions of the country tell about dry weather caused by lack of rain and excessive heat. According to one source no rain fell between early May and late September, except for some thunder rains in Karlstad and at Letstigen in the province of Närke in June. Grain and hay harvests suffered severely except for rye and particularly in Finland, which fared slightly better. Great bushfires were rampant, destroying forests and rye in the fields. Watermills stood still due to dried out rivers. The heat caused epidemics killing many people, including members of the Royal Council. Also, from 1657 there are reports covering all of Sweden about severe drought. Already in April the gardens were "longing for rain". In Johan Rosenhane's diary from Östergötland every day is noted to have been hot or very hot weather from early May to late August. Both the month of August and the entire year is said to have been so dry and hot that wells and streams went dry in Småland and Östergötland and that no one could remember such a drought. In the spring, eleven out of 65 iron mills in the Bergslagen region were unable to operate due to lack of water, especially those located by smaller rivers, and most of them had to limit their operations considerably during the whole year. The lack of water in the rivers running into Lake Mälaren is also shown by the fact that the water level of the lake was so low that sandbanks were visible. Even in the northern province of Norrbotten the summer drought caused forest fires and much damage on the harvest.

3) 1665-1670. The last years of the 1660s was a new period of dry years. 1666 seems to have been the worst; already in July harvests were forecasted to fail and at least in the west there was a lack of rain between late June and late September. But also in all of the following four years harvest failures are reported and water levels in lakes and streams were extremely low.

4) 1677-1684. The same pattern was repeated in the end of the 1670s and early 1680s. In particular, 1681 and 1684 stand out; in the former year Stockholm had no rain at all in April and May and hay harvests were weak, and in 1684 there was a food crisis, the peasants requiring to pay their church tithes in cash rather than in grains.

5) 1746-1750. A new prolonged drought period occurred in the mid-1700s. Beginning in 1746, there are repeated reports on spring drought, and in the following years also summer drought from Hälsingland in the north to Västergötland in the west. Streams dried up and harvests failed and bark beetles, favoured by the hot weather, destroyed timber wood.

6) 1757-1767. Most of the growing seasons of this period were affected by dry weather with harvest failures and dried up wells and marshes. Spring was particularly late in 1758; in the Stockholm harbor ice was said to be one meter thick in late April and there was still ice in inlets and small lakes in early May. The following summer was hot and dry, as were the summers of 1759, 1762 and 1764. According to one source, the dry period extended from 1749 to 1767 at least in the north and with annually varying degrees of intensity.

7) 1771-1776. According to sources covering most of the southern half of the country these years were all characterized by cold springs and hot and dry summers. Hay harvests failed due to dried up wet meadows and even rye failed to mature in due time. In particular 1775 stand out as a critical year. Barley, peas and hay suffered severely and lake water levels reached record lows. In the Stockholm region famine threatened in 1771.

8) 1780-1783. From Västerbotten in the north to Blekinge in the south there are reports on cold springs and dry summers, dried-up wells and streams, bushfires, and in Västergötland marshes were even so dry that they caught fire. In 1782, sowing was delayed until the first week of May in the Stockholm region due to persisting ground frost. In Västerbotten in the north it only rained twice from summer to October in 1780 and roots and cabbage failed, while the rye harvests were quite good as was the hay harvest, probably due to cultivation on wet meadows watered by meltwater from the mountains. On the other hand, in all regions in the south the hay harvest seems to have failed and the price of rye rose with more than a third over the year. The same pattern was repeated in 1781 and 1783.

*Results*

Figure 2 shows a scatter plot between summer temperatures and the drought index. The correlation from Table 1 of 0.47 is expressed as $R^2$ in Figure 2. This might be a consequence of the precipitation data prior to 1893 not being very good. Another possibility is of course that the drought index really is more of a JJA temperature index.

Fig. 2: Scatter plot of JJA temperature 1756-1816 and the drought index for Sweden

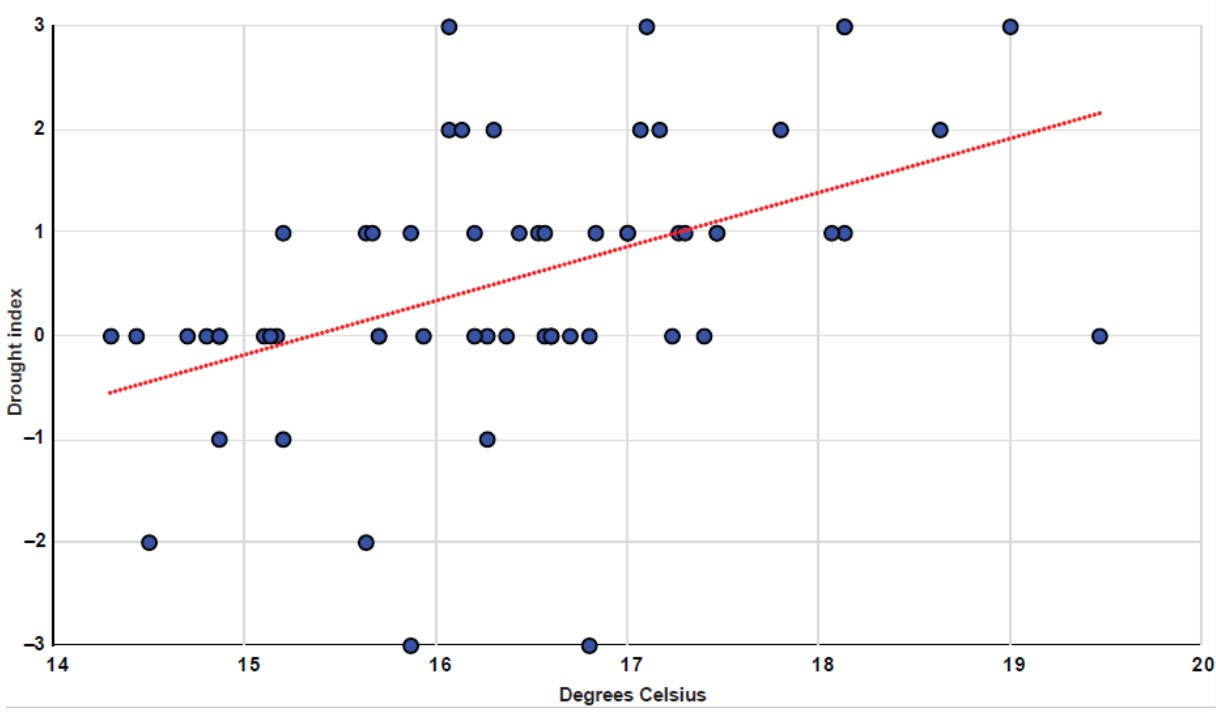

Note that "Very dry" is +3, which should correspond to low levels of precipitation.

Another difference to precipitation data is that the index hardly has any correlation with August temperatures, while instrumental data renders a (slight) correlation between temperature and precipitation in August. We believe the main reason for this might be that the

database may be more stringent when it comes to weather related events occurring during the first half of the year. It is also possible that a cool May, may be experienced as "wet", and therefore described as such in the sources forming the foundation of the index.

These tentative results of comparing the drought index made from descriptions of droughts and precipitation indicate that the descriptive sources are indeed correlated to climatic variables of temperatures and precipitation. Also, although correlation is higher between temperature and index, than between precipitation and index, the original data concern descriptions of dry or wet conditions: i.e. a description like "a hot/warm summer" is not included in the index.

Since we have temperature measurements for the latter half of the 18[th] century, it is possible to quantify periods 6, 7 and 8 (1757-1767, 1771-1776 and 1780-1783) in the section above. In Table 2, average monthly temperature for June, July and August, as well as the summer season JJA, are compared to average monthly temperature for the entire period 1756-1816, i.e. until that year the index ends. None of the dry sub-periods differ significantly from average monthly temperature for any of the summer months, or of the summer season. The period of 1771-1776 has the highest difference compared to average monthly temperature for the whole period 1756-1816, being c. 1 degree C warmer.

Table 2: Dry periods in the 2[nd] half of the 18[th] century in Sweden reflected in instrumental measurements. Average monthly temperature for 3 sub-periods

| Period | June | July | Aug | JJA |
|---|---|---|---|---|
| **1756-1816** | 14.88 | 17.81 | 16.47 | 16.39 |
| **(Index period)** | (1.62) | (1.61) | (1.51) | (1.17) |
| **1757-1767** | 15.69 | 17.99 | 16.19 | 16.62 |
| | (1.44) | (1.59) | (1.18) | (0.63) |
| **1771-1776** | 16.50 | 18.95 | 17.13 | 17.53 |
| | (1.44) | (1.16) | (1.68) | (1.13) |
| **1780-1783** | 15.63 | 18.58 | 17.53 | 17.24 |
| | (1.58) | (2.25) | (2.02) | (1.56) |

From Figure 2 it is visible that when average JJA temperature is 17 degrees C or higher, there are no indications of excessive precipitation. Droughts, on the other hand, are prevalent from +15 degrees C, and very dry conditions may occur if temperatures are above 16 degrees C, confirming the average temperatures in Table 2.

*Discussion and conclusions*

In this paper we tried to show that turning descriptions of drought (and precipitation) into an index do correlate with instrumental measures of drought and temperature. We also provided descriptions of periods that suffered harvest failures through drought, precipitation as well as some adverse temperatures for the 15[th] century. Since the data are so scarce for the period, we have not included 13[th] to 15[th] centuries into the index. Even results concerning, at least the first half of, the 16[th] century ought to be regarded as uncertain.

The main problem with the precipitation/drought index is that we have a very short period (1786-1816) with overlapping data of precipitation/drought. Also, instrumental data on precipitation might not be of very high quality. Therefore, lack of any correlation between the index and precipitation data may have three reasons. 1) The index rather reflects summer temperatures than drought/precipitation. 2) The instrumental precipitation data for the late 18[th] and early 19[th] century is not of very high quality. There is some correlation between the drought index and summer temperatures in Stockholm, just like there is some correlation

between precipitation and summer temperatures. Correlation between the drought index and summer temperatures is higher than between summer temperatures and precipitation, so it is possible that the drought index is rather a temperature-index. Hot summer temperatures will cause drought, because in Sweden, it very seldom rains when the weather is hot. 3) The drought index reflects data that come from different parts of Sweden. Instrumental precipitation data are, of course, from a very limited geographical area and will not reflect a general drought in Sweden.

Despite the shortcomings of the index, we still think that some conclusions may be drawn from it.

First: the height of the Little Ice Age, between c. 1570-1630, is, characterized by very high variations with some years extremely wet, and some years extremely dry.

Secondly, after the early 1660s, wet years became increasingly uncommon, and most years are either dry or very dry, especially from the mid-1700s onwards. Although previous estimates of Stockholm temperatures after 1756 have showed to be positively biased, this seems to correspond to trends in TRW and density in at least northern Fennoscandia (Moberg et al 2003, Grudd 2008).

For the late 13$^{th}$ to the early 16$^{th}$ century, lack of data has made it impossible to extend the index so far back in time. Grain prices suggest difficulties for grain production around the turn of the century 1300. The highest price ever might reflect the catastrophic years of 1314-16 – but the harvest failed that year because of wet and cold (Slavin, 2018: 495-515) Therefore, we argue that just grain prices cannot determine a specific climatic parameter (at least not for Sweden), since different conditions (too wet or too dry) result in the same outcome (dearth and higher prices).

Since the index is a made of discrete variables, we thought it less meaningful to try out a regression analysis and model (which would only render 7 different "temperatures"), especially since we have been concentrating on precipitation and not temperature. Finally, Table 3 summarizes the index presented in Figure 1:

Table 3. Number of years that have been labelled anything but "normal"

| Index number | -3 | -2 | -1 | 1 | 2 | 3 |
|---|---|---|---|---|---|---|
| Number of years | 13 | 19 | 18 | 69 | 24 | 19 |

Table 3 indicates that slightly dry (index value +1) years have been regarded as more "exceptional", than wetter (-1) years. When it comes to very wet (-2) or very dry (+2) and exceptional wet (-3) or exceptional dry (+3) years, there are a few more years denoted as dry than as wet. Out of 316 years, only 13 years were exceptionally wet and 19 were exceptionally dry. Additional 19 years were very wet and 24 were very dry. That so few years, comparably, were regarded as "wet" years (only 18) compared to 69 that were regarded as dry years, may be a result of perception: nice summers will be commented upon.

**Archival sources**

Riksarkivet (National Archives of Sweden), Stockholm, Medieval codex A21
Riksarkivet (National Archives of Sweden), Stockholm, Sturearkivet

Riksarkivet (National Archives of Sweden), Stockholm, Brev från Catharina Wallenstedt, 1627-1719. Brev till dottern Margareta och sonen Carl. RA, Sjöholmsarkivet 1 enskilda samlingar, Ehrensteens samling, vol 2

Riksarkivet (National Archives of Sweden), Stockholm, Landshövdingars skrivelse t K M:t, Jönköpings län, Östergötlands län, Södermanlands län, Uppsala län, Stockholms län

Riksarkivet (National Archives of Sweden), Stockholm, Kollegiers m fl skrivelser t K M:t. Generalguvernörers skrivelser, generalguvernören över Skåne, Halland samt Göteborgs- och Bohus län

Stockholms stadsarkiv (City archives of Stockholm), Magistratens ämbets- och byggnings Kollegium, Slussverket. Wattu journal 1774–1819

**Temperature and precipitation datasets**

The datasets are freely available and were downloaded from Bolin centre https://bolin.su.se/data/stockholm-historical-temps-monthly on 7th December 2019 and from SMHI: https://www.smhi.se/data/meteorologi/ladda-ner-meteorologiska-observationer/#param=precipitationMonthlySum,stations=all,stationid=98210 on 5th February 2020.

E-mail contact with SMHI confirmed that precipitation data from 1863 are missing.

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

Table 3: Documentary evidence of droughts in Sweden 1600-1800

**1634-1639**

| Year | Date | Location | Index | Source | Comment |
|---|---|---|---|---|---|
| 1634 | spring, summer, autumn | [Sweden], Västergötland, Norrland | 2 | Falkengren 1781; Bergh 1886: 194; Bergh 1888: 56; Wittrock 1948; Sondén 1890: 363; Edén 1905: 216 | dry April, dry and hot summer, harvest failures, great drought and hailstorms, lack of water, poverty |
| 1635 | summer | [Sweden] | 2 | Falkengren 1781; Edén 1905: 216 | great drought, lack of water, bad hay harvest |
| 1636 | spring, summer | [Sweden] | 2 | Falkengren 1781 | dry spring, hot and dry summer, no rain in May and only little before June 13 |
| 1638 | spring, summer | Dalarna, Stockholm | 1 | Falkengren 1781; Norberg 1958-1959: 23 | drought, lack of water, dry late spring and early summer |
| 1639 | spring, summer | Värmland, [Sweden] | 2 | Löf 1942: 151; Falkengren 1781 | dry spring, June hot and dry, harvest failure in Värmland due to persistent drought |


**1652-1657**

| Year | Date | Location | Index | Source | Comment |
|---|---|---|---|---|---|
| 1652 | summer | [Sweden, Finland] | 3 | Hausen 1880: 183, 198-201; Ambrosiani 1923: 255; Malmberg 1917: 87; Rääf 1856: 349; Hannerberg 1941: 206; Sillén 1855: 103; Ahlqvist 1825: 295; Weibull 1923: 114 | Great drought; no rain in Sweden or Finland and forest fires between early May and late September, except for 25 and 30 June in Närke and Karlstad, Great harvest failure for both grain and hay, although somewhat better for rye, lack of water in streams |
| 1655 | July | Värmland | 1 | Hausen 1880: 219 | Dry weather all July |
| 1657 | summer | Västergötland, Östergötland, Västerbotten, Västmanland | 3 | Sjöberg 1915: 21; Jansson 1995; Ambrosiani 1923: 256; Weinhagen 1947: 68; Isacson 2004: 130; Göthe 1929: 119; Hülphers Abramsson 1793: 318; Steckzén 1981: 77 | Hot and very dry |


**1665-1670**

| Year | Date | Location | Index | Source | Comment |
|---|---|---|---|---|---|
| 1665 | summer | Stockholm, Småland | 2 | Fryxell 1836: 137-138; Thunaeus 1968: 252 | Strong heat, dead fish, great city fires due to drought |
| 1666 | summer, autumn | Värmland, Blekinge, Västergötland, Halland | 3 | Hausen 1880: 338, 340-2; Petersson 1942: 66; Landshövdingen öfver Skaraborgs län | drought, grain and grass die, low water in lakes and streams, watermills stand still due to lack of |

| | | | | | |
|---|---|---|---|---|---|
| | | | | Tord Bonde Ulfssons berättelser för åren 1661-1666: 144; Osbeck 1922: 18; Ahlqvist 1825: 295 | water, forest fires, cabbage hit by worms due to the drought, cattle disease, great poverty, in Halland no rain between midsummer and late September |
| 1667 | spring, summer | Värmland, Östergötland | 2 | Hausen 1880: 363; Rääf 1856: 349; Westerlund and Setterdahl 1917: 6 | cold and dry spring, dry summer, general harvest failure |
| 1668 | spring, summer | Västergötland, Norrbotten, Östergötland | 1 | Tilander 1976: 186; Olofsson 1974: 227; Rääf 1856: 349 | dry spring, harvest failures |
| 1669 | spring | Östergötland | 1 | Rääf 1856: 349 | dry spring |
| 1670 | spring | Dalarna | 1 | Lindroth 1955: 157 | drought, watermills stand still |

**1677-1684**

| Year | Date | Location | Index | Source | Comment |
|---|---|---|---|---|---|
| 1677 | autumn | Västerbotten, Uppland, Västergötland | 1 | Nordlander 1938: 115; Sjöberg 1976: 35; Bergstrand 1955: 36 | great drought, harvest failure, low water in streams, watermills stand still |
| 1678 | summer | Dalarna, Västergötland | 2 | Söderberg 1999: 110; Bergstrand 1955: 37 | Tiny grain due to drought, harvest failures |
| 1679 | summer | Uppland, Småland, Öland, Skåne, [Sweden] | 1 | Jansson 1947: 74-5; Brunnström 1913: 78-9; Hegardt 1975: 144; Fredriksson 1979: 175 | low water in streams, some watermills standing still half a year, no rain in southeastern Småland and Öland between midsummer and 25 July, great drought and harvest failures |
| 1680 | summer | Uppland | 1 | Jansson 1947: 74-5 | watermills stand still for 11 weeks due to lack of water in the streams |
| 1681 | spring, summer | Stockholm, Södermanland | 3 | Wijkmark 1995: 246, 265; RA Brev från Catharina Wallenstedt 4 May and 30 June 1681; Levander 1934: 37 | unprecedented heat in April, no rain for 8 weeks and much heat in May and June, bad hay harvest, people eat bark bread |
| 1683 | spring | Gästrikland, Skåne | 1 | Norberg 1958-1959: 376; RA Kollegiers m fl skrivelser t. K. M:t Generalguvernören över Skåne, Halland samt Göteborgs och Bohus län 11 July 1684 | watermills stand still since September 1682 due to lack of water in streams, harvest failures |
| 1684 | summer | Östergötland, Småland, Skåne, | 3 | RA Landshövdingens i Östergötlands län skrivelse till K. M:t 20 June 1684; RA Landshövdingens i Jönköpings län skrivelse till K. M:t 9 | great drought, harvest failures, poverty, grain price increases, watermills stand still due to lack of water in streams |

| | | | | July 1684; RA Landshövdingens i Södermanlands län skrivelse till K. M:t 13 October 1684; RA Kollegiers m fl skrivelser t. K. M:t Generalguvernören över Skåne, Halland samt Göteborgs och Bohus län 21 and 28 July, 21 August, 15 September, 6 October 1684; Omberg 1992: 46 | |

**1746-1750**

| Year | Date | Location | Index | Source | Comment |
|---|---|---|---|---|---|
| 1746 | spring, summer | Medelpad, Västergötland, Uppland, Hälsingland, [Sweden] | 3 | Nordenström 1894: 43; Utterström 1957: 430; Olander 1951: 119; Pehrsson 1781; Lindgren 1971: 127; Hiorter 1747; Broman 1911: 524; | heat and drought before midsummer, forest fires, harvest failures particularly for grain |
| 1747 | spring, summer | Hälsingland, Uppland, Västergötland, [Sweden] | 1 | Broman 1911: 530; Utterström 1957: 109, 431; Lindgren 1971: 127; Wallén 1910: 3, 13; Olander 1951: 119; Pehrsson 1781; Fritz 2010: 68 | severe drought with no rain in all of May, drought in July, August and September, poor grain and flax harvest in Hälsingland, great harvest failure on grain, and bark beetles proliferate in spruce forests, low water in lakes and streams |
| 1748 | spring, summer | Västergötland, Uppland, Östergötland, Hälsingland, Medelpad, Småland, [Sweden] | 3 | Trolle-Bonde 1894: 149; Elvius 1748: 39, 53-4; Hiorter 1752: 101-9; Nordenström 1894: 44; Hofrén 1984: 296-7; Broman 1911: 530; Olander 1951: 119; Utterström 1957: 109; Palm 1997: 134; Wallerius 1779; Lindgren 1971: 127; Ilmoni 1853: 127; Trolle-Bonde 1894: 149; Ejdestam 1969: 77-9; Wallén 1910: 3; Fritz 2010: 68 | heat and drought, only little rain in July, low water in lakes and streams, hay harvest reduced to 25-33% in relation to the previous year in Västergötland, only little rain May-September in Stockholm, |
| 1749 | spring, summer | Uppland, Medelpad, Hälsingland, Östergötland, Närke | 1 | Utterström 1957: 109; Fritz 2010: 68; Nordenström 1894: 42, 44; Osvald 1965: 68; Hannerberg 1941: 215 | heat and drought in the spring, low water in lakes and streams, only little rain in the north May-September, bad potato harvest |

| Year | Date | Location | Index | Source | Comment |
|---|---|---|---|---|---|
| 1750 | summer | Medelpad, Västmanland, Uppland, Hälsingland | 1 | Nordenström 1894: 42; Omberg 1992: 50; Utterström 1957: 431; Schissler 1972: 52 | very hot and dry summer, only little rain in the summer and low water in the streams in Medelpad and Västmanland, bad hay harvest |


**1757-1767**

| Year | Date | Location | Index | Source | Comment |
|---|---|---|---|---|---|
| 1757 | summer | Västergötland, Medelpad, Skåne, Halland, Öland | 2 | Pehrsson 1781; Nordenström 1894: 45; Ejdestam 1969: 77-9; Wallén 1910: 14; Osbeck 1922: 17; Ahlqvist 1825: 295; | hot and dry summer, harvest failures, June-August no rain in Skåne, low water levels in lakes, marshes dried up, |
| 1758 | spring, ummer | Södermanland, Värmland, Västergötland, Östergötland, Halland, | 2 | Tessin 1819: 334; Hellgren 1996; Pehrsson 1781; Widegren 1828: 449; Wallén 1910: 14 | dry spring, summer and autumn, tiny grain harvest, low water levels in lakes |
| 1759 | spring, summer | Halland, Västergötland, Uppland | 0 | Pehrsson 1781; Wallerius 1779; Osbeck 1922: 17; | dry spring and hot summer |
| 1761 | spring, summer | Uppland, Södermanland | 1 | Ejdestam 1969: 77-9; Tessin 1819: 358 | drought spring and summer, in many places grain harvest failures |
| 1762 | summer | Medelpad, Västergötland, Småland | 3 | Nordenström 1894: 45; Pehrsson 1781; Ejdestam 1969: 77-9; Sidenbladh 1908: 94 | only little rain in Medelpad in July, severe drought in June and July in Småland and before midsummer in Västergötland but rain in the autumn, severe drought and bad hay and grain harvest in Uppland |
| 1763 | summer | Medelpad, Uppland | 0 | Nordenström 1894: 45; Sidenbladh 1908: 94 | drought in June, bad hay and grain harvest in Uppland |
| 1764 | spring, summer | Medelpad | 1 | Nordenström 1894: 45-6; Sidenbladh 1908: 94; Wallerius 1779 | tiny harvest due to cold spring and dry summer, severe drought and bad hay and grain harvest in Uppland |
| 1765 | summer | Medelpad | 1 | Nordenström 1894: 46 | dry fields and northern winds, lack of food |
| 1766 | summer | Medelpad, Västergötland | 1 | Nordenström 1894: 46; Pehrsson 1781 | drought and worms destroyed the grain harvest, July dry in Västergötland |
| 1767 | summer | Medelpad | 1 | Nordenström 1894: 46 | drought, rain in mid-July could not be absorbed by the dry soil |


**1771-1776**

| Year | Date | Location | Index | Source | Comment |
|---|---|---|---|---|---|
| 1771 | spring, summer | Uppland, Stockholm, Värmland, Östergötland | 2 | RA Landshövdingens i Uppsala län skrivelse till K Maj:t 18 June 1771; Landshövdingens i | cold spring, severe and protracted drought in early summer and then rain, harvest failures |

| Year | Date | Location | Index | Source | Comment |
|---|---|---|---|---|---|
|  |  |  |  | Stockholms län skrivelse till K Maj:t 12 October 1771; Ejdestam 1969: 77-9 |  |
| 1772 | spring, summer | Uppland, Östergötland, Västergötland | 2 | Wallerius 1779; Hushållnings Journal October 1786; Ejdestam 1969: 77-9 | dry spring and early summer, drought and harvest failures, widespread hunger |
| 1773 | summer | Halland | 1 | Barchaeus 1924: 97 | severe drought in July, harvest failure |
| 1774 | summer, autumn | Stockholm, Östergötland | 3 | Stockholms stadsarkiv, Magistratens ämbets- och byggnings Kollegium, Slussverket. Wattu journal 1774; Hushållnings Journal October 1786 | severe drought early July to mid-October, bad grain harvest |
| 1775 | summer | Östergötland, Uppland, Värmland, Västergötland | 3 | Hushållnings Journal October 1786; Anteckningar ur Statistiska tabeller för Stockholms-Näs, 1749-1859; Wallerius 1779; Danielson 1974: 37; Schiller 1933: 340-1 | dry spring and severely hot summer, bad harvests of hay, peas and grain, high grain prices, low water levels in lakes |
| 1776 | summer | Västergötland | 1 | Schiller 1933: 341 | bad hay and grain harvests, low water levels in lakes |


**1780-1783**

| Year | Date | Location | Index | Source | Comment |
|---|---|---|---|---|---|
| 1780 | spring, summer, autumn | Hälsingland, Stockholm, Blekinge, Västergötland, Västerbotten, Öland, Dalarna, Skåne | 3 | Ny journal uti hushållningen 1776-1813, del 1: 159, 172, 174, 176-7, 232-3; Stockholms stadsarkiv, Magistratens ämbets- och byggnings Kollegium, Slussverket. Wattu journal 1780; Schiller 1933: 342 | dry spring and summer, low water levels in lakes, streams and wells, extreme drought in the autumn, bushfires, few bees |
| 1781 | spring, summer | Stockholm, Värmland, Östergötland, Västernorrland, Västergötland, Västerbotten, Småland, Öland | 3 | Ny journal uti hushållningen 1776-1813, del 3: 192, 235-6; Utterström 1957: 435; Stockholms stadsarkiv, Magistratens ämbets- och byggnings Kollegium, Slussverket. Wattu journal 1781; Hushållnings Journal October 1786; Schiller 1933: 342-3; Åmark 1915: 238; Bergstrand 1954: 40-1 | repeated drought periods April to September, harvest failures particularly for hay, low water levels in lakes, forest fires |

| 1782 | spring, summer | Stockholm | 0 | Ny journal uti hushållningen 1776-1813, del 3: 239 | cold and dry spring |
|---|---|---|---|---|---|
| 1783 | spring, summer, autumn | Stockholm, Halland, Östergötland, Uppland, Halland [Sweden] | 2 | Ny journal uti hushållningen 1776-1813, del 3: 234, 243-5; Utterström 1957: 436; Stockholms stadsarkiv, Magistratens ämbets- och byggnings Kollegium, Slussverket. Wattu journal 1783; Anteckningar ur Statistiska tabeller för Stockholms-Näs, 1749-1859; Osbeck 1922: 17; | cold and dry spring, protracted summer drought, harvest failures, low water levels in streams in the autumn |
