# Peer review of "DOCUMENTARY EVIDENCE OF DROUGHTS IN SWEDEN BETWEEN THE MIDDLE AGES AND c1800"

_Climate of the Past, 2020_

## Referee Comment (RC1) · Anonymous Referee #1 · 8 May 2020

This article is about droughts in Sweden from 1400 to 1800 reconstructed on the basis of documentary data. Besides the reconstruction of droughts, it also deals with their economic consequences and presents a number of drought periods. It is a very interesting paper and presents important results, which are definitely worth publishing. Nevertheless, there are some points of criticism that would have to be changed before publication: Actually, a proper state of research is missing. Some of this information appears in the course of the paper, but it would be better to concentrate it at the beginning (e.g. lines 166-167). At the beginning of the paper, but also further down, there are also a few references missing. For example, the description of the climate indices (lines 77-79) should be followed by a reference to Christian Pfister or Rudolf Brázdil, who developed them in this form. Or lines 156-159, this statement should also be fol-

lowed by a reference. On the other hand, I think the table with the references for lines 290-340 is an excellent solution! In my opinion, the introduction also lacks a delimitation of the subject or a kind of outline. Therefore, a genuine question or something similar, which defines the framework of the article, would be very helpful here. It should also be considered whether the description of the data and historical sources should possibly be separated from the introduction and presented in a separate chapter. If so, more information on the nature of the database could be added in this chapter as well as information on grain prices in the sources (lines 163-164). I agree in principle with the statement in lines 175-177, but nevertheless this statement that grain prices in the 15th century were mainly influenced by meteorological events should be better substantiated (e.g. references!). It should also be borne in mind that the demand and price relationships between the various cereals can fluctuate, even if the basic demand for cereals remains generally stable. Furthermore, I would be very careful to associate the famines of the 15th and early 16th centuries described above with drought. In some examples this is plausible, in other examples less so. Many famines in this period coincide with price increases in other parts of Europe and the reasons for these famines there were often different weather patterns than droughts. Of course, the article does not say this explicitly, but the statement in lines 156-159 already suggests it. However, I would like to emphasize again that this article contains valuable results and I recommend a publication after minor changes.

Minor remarks: In the abstract the listing of drought periods is not chronological. This may be the intention, but then it would have to be written slightly differently. Fig 1. has a somewhat unusual scale as a label for the x-axis. Would intervals of 10 years perhaps be somewhat clearer? The years in the graph are rather difficult to read anyway. lines 107-111: Maybe the links and especially the information about the last access etc. would fit better into the reference list.

---

## Referee Comment (RC2) · Anonymous Referee #2 · 17 May 2020

The manuscript is an interesting attempt to discuss how historical documentary evidence can be used to reconstruct droughts in Sweden between ca 1400 and 1800. This is, to the best of my knowledge, something that has not been undertaken before and so it is a welcome contribution to the field of historical climatology. Parts of the discussion is well written, in particular the part between line 163 and 275, where details in what the historical documents can tell us are presented. As a whole, however, the article has major problems that must be addressed before the evidence discussed can be acceptably presented. Thus, my overall recommendation is that the manuscript needs major revision.

Below, I will outline the most important aspects at large. I will not comment much on various minor things because I believe that the manuscript after a major revision will

necessarily look quite different in its details.

Method

I do not find the method description satisfactory. Essentially all that is said is a vague mentioning of a 7-point index scale ranging from -3 (extremely wet) to +3 (extremely dry). Nothing is said of how the authors have made in order to decide which value on the scale to choose for a particular year. Because their method is not clearly described, it is impossible to understand how they constructed their index time series. A clear method description is needed, preferably preceded by a discussion of relevant aspects (both strengths and weaknesses) and methodological problems encountered in similar investigations made by other scholars (in other countries) in the past.

The authors state on lines 81-82 that they used an already existing index of wet years (from Retsö 2015), which they complemented with indexed notes on drought. Nothing, however, is said of how the existing index and the new information was combined. This must be made clear.

Because it is relevant here, I also read the Retsö 2015 paper and I note that the table in Appendix A there presents much clearer information, where the reader at least can obtain a direct view of what kind of statements in the historical documents that lie behind the choice of a particular value on the index scale. Something similar here should help readers of the new work to understand it better, even without a strictly defined methodological rule.

I find it confusing that the authors refer to their constructed index several (seven) times as a "precipitation index" instead of a drought index, which one would excpect that they should call it given how they start their account. Is it a drought index, or is it a precipitation index?

Documentary data

There is insufficient information in the manuscript regarding which documentary evidence that were used. There are indeed some archival sources mentioned and the reference list contains several elements. But it is anyway impossible to understand which the actual underlying data are.

At four instnances the authors refer to "the database"; on line 12 (in the abstract), 85 (section 'Method'), 137 (section 'Instrumental measurements'), and 363 (section 'Discussion and conclusions'). However, they provide no information of where this "database" can be found or how it has been constructed. This is not acceptable and it does not match the Data Policy of the journal, which says that "it is particularly important that data and other information underpinning the research findings are "findable, accessible, interoperable, and reusable" (FAIR) not only for humans but also for machines." In order to comply with this policy, the authors are advised to make their database, including sufficient metadata to understand the actual data, available to the public in a recommendable digital repository for research data, and provided with a persistent identifier (digital object identifier).

Instrumental data

The authors use early instrumental temperature data from Stockholm in order to calculate correlation coefficients with their index series. This is fine and it illustrates that their index series has some correlation with summer temperatures. However, they say twice (lines 141 and 357) that they have no overlapping precipitation data. This is surprising given that they have used more recent precipitation data from Stockholm starting in 1859, which they have downloaded from the SMHI Open Data reseource. Maybe the authors simply are not aware of it, but the same SMHI resource that they already have used actually also contain freely available digital monthly precipitation data from relevant stations that do overlap in time with their own index series. The following station precipitation data are available:

Uppsala 1723-1732 and 1739-present, Stockholm 1786-present, Risinge 1730-1740.

Even if early precipitation data certainly may have their own problems, it would be

much better to use them - in particular the long Uppsala record - to calculate correlation coefficients with the index series than not doing this. It would also be advisable here to refer to the recently published global inventory of pre-1850 instrumental meteorological records by Brönniman et al. (2020) (https://journals.ametsoc.org/doi/10.1175/BAMS-D-19-0040.1). The catalogue in the supplementary material to that paper does include the above mentioned precipitation data from Uppsala, Stockholm and Risinge.

I recommend the authors to use the available overlapping monthly precipitation data in order to study correlations with their index data. Once this has been done, it may have a noticable impact on how they discuss and interpret their index series.

Statistical analysis

The authors calculate correlation coefficients between instrumental temperature data and their index series, and between instrumental and precipitation data. However, this part of their analysis lacks a clear motivation, a clearly stated goal, and also statistical rigour. The authors are advised to be more stringent and apply appropriately chosen siginficance tests and use the subsequent result in their discussion.

The content of the tables with correlation coefficients is difficult to understand and I find it superfluously accurate to present three decimals in the calculated coefficients. Two decimals are enough in this context.

Figures

All figures are unfortunately rather poorly produced. Please provide more professional-looking figures.

The content of Figure 3 is confusing. Why are data before 1400 presented there at all? And which are these data? If they are needed in the discussion, this should be argued for and references to the data should be given.

Discussion and conclusions

[Figure]

There are several issues that need to be better discussed. I will not mention everything I can think of here because I believe that the discussion after a major revision naturally will be quite different.

I find, however, the following statement (lines 360-363) particularly noteworthy:

"Correlation between the precipitation index and summer temperatures are higher than between summer temperatures and precipitation, so it is possible that the precipitation index is rather a "good-summer-weather-index". We think that the ideal would be to extend the documentary database until – at least – early 20 th century."

If the index series is a "good-summer-weather-index" rather than a precipitation index (or a drought index, which I think is what the authors should call it), then how much scientific information value do the conclusions that emerge from this paper have? In particular, how should one understand the claimed time periods of droughts that are discussed and even highlighed in the abstract? Are they periods of drought or periods of "good" summer weather? If their interpretation is that uncertain, should one really present them as periods "particularly struck by summer droughts" in the abstract?

Why do the authors "think that the ideal would be to extend the documentary database until – at least – early 20th century." It would be interesting to hear some discussion about what such an extension could help us with.

Given that the authors seem to not be convinced themselves that their derived index really shows us when droughts occurred, it would - overall - be advisable that they put more efforts on discussing the question of whether it is possible to develop a reliable drought reconstruction from Swedish documentary data.

As a last remark, I should say that I do encourage the authors to re-consider how to approach the documentary data and how to present and discuss them. The topic of drought reconstruction is clearly of interest, not the least in light of the fact that Sweden only two years ago (2018) experienced an exceptionally warm/dry summer. Scientific

papers about the 2018 summer in Sweden and nearby have recently started to appear in the climatological literature. It could be a good starting point in a revised discussion and analysis of the here presented material, to relate it to what meteorologists and other climate scientists recently have considered be of importance in this context. Historical documentary data have the potential to tell us whether similar weather conditions have happened anytime during the past five hundred years or so.
* * *

---

## Author Comment (AC1) · 1 Jul 2020

Answers to 'Interactive comment on "Documentary evidence of droughts in Sweden between the Middle Ages and c 1800"'. #Referew1 State of the art: We acknowledge the referee's view that a proper "State of the art" at the beginning of the paper. However, we preferred to keep a natural discussion though out the paper. #Referee 1 : "References missing" & Referee 2:" Method". We agree with both referees and have tried to develop the discussion of method. Referee 2 also criticise our statistical analysis. We agree with that too, and propose a much shorter version, since we came to realize that the many correlations presented in the first version did not say anything of particular value. However, we argue that a proper regression analysis to the material would be – as it stands now and with the present available data – to overreach the explanatory

capacities of the original data. #Referee 2 do not like "precipitation index" in a paper of drought. We have used this term since most data considered have been about too little rain causing the drought. #Referee 1 suggests a genuine question to the paper, which we have provided in the new version: "Is it possible to distinguish periods of drought in Sweden through documentary sources from the 15th till the 18th century?" Both Referee 1 & 2 asked for more details on the database, which we have tried to present in a better and more comprehensive way – where it comes from, how it was created and that it will soon be publicly available at the Bolin Centre for Climate Research. However, the database is in Swedish. We can't see how that could be changed, since the documents underlying the excerpts of the database are in Swedish. Should we translate the database? Translations are very tricky and misunderstandings and anachronisms would be bound to occur. Historical descriptions and data are of course bound within their historical, political, religious, legal and geographical context. Referee2 was very critical to our figures. Most of them have been taken away. Referee 1 criticise Fig 1 for odd scales (also that it ranged from -4 to 4, which we found very odd, since our own figure ranged from -3 to 3). We have tried to fix this. Referee 2 is not satisfied with our discussion and that we are too hesitant in describing the index as a drought index (and not as a "good-summer-weather-index"). We have reworked the paper and try to show that although correlation is higher between temperature and precipitation, the original data used ARE descriptions of drought or wetness rather than descriptions of hot or cold. But in Sweden, high temperatures results in dryness and cold temperatures in wetness. We are very grateful to both referees and they have much improved the paper. Yours truly, Lotta Leijonhufvud Dag Retsö
* * *

---

## Author Comment (AC2) · 6 Jul 2020

State of the art: We acknowledge the referee's view that a proper "State of the art" at the beginning of the paper. However, we preferred to keep a natural discussion though out the paper. "References missing". We agree and have tried to develop the discussion of method. The referee a genuine question to/delimitation of the paper, which we have provided in the new version: "Is it possible to distinguish periods of drought in Sweden through documentary sources from the 15th till the 18th century?" The referee asked for more details on the database, which we have tried to present in a better and more comprehensive way – where it comes from, how it was created and that it will soon be publicly available at the Bolin Centre for Climate Research. However, the database is in Swedish. We can't see how that could be changed, since the

documents underlying the excerpts of the database are in Swedish. Should we translate the database? Translations are very tricky and misunderstandings and anachronisms would be bound to occur. Historical descriptions and data are of course bound within their historical, political, religious, legal and geographical context. Translation of a database of more than 20,000 excerpts is out of limit for the authors. When it comes to famines and droughts in the 15th and 16th century, we do not associate famines with droughts. We try to make it clear that agricultural production does have some climatic signal, but it is very difficult – if at all – to assess the signal. We all know that grain production to a very large extent is susceptible to both drought and excessive rain: two different phenomenon, same result. We hope to make this clear in the text. However, we find the grain price data interesting and would like to include them, since they clearly depict some climatic variations. Also, grain prices are the longest series we have, and the catastrophic (rain) year of 1315 also shows up in the Swedish material. The referee criticises Fig 1 for odd scales (also that it ranged from -4 to 4, which we too found very odd, since our own figure ranged from -3 to 3). We have tried to fix this, as well as the other minor changes concerning not chronological drought periods. We are very grateful to the referee who has much improved the paper. Yours truly, Lotta Leijonhufvud Dag Retsö

———————————————————

---

## Author Comment (AC3) · 6 Jul 2020

" Method". We agree with the referee and have tried to develop the discussion of method, which in essence consists of close reading, evaluation and counting of mentions of different words for "dry", "drought" (and precipitation) that made us conclude that a year had been unusually dry. Referee 2 also criticise our statistical analysis. We agree with that too, and propose a much shorter version, since we came to realize that the many correlations presented in the first version did not say much of particular value – and simply excluded most of the analysis of instrumental values – Moberg et al have already done that much better than we ever could. We argue that a proper regression analysis to the material would be – as it stands now and with the present available data – to overreach the explanatory capacities of the original data. As for

including data from different stations – yes, that might have been more appropriate. However, merging data from different stations into a composite precipitation series, in combination with different lengths of the series from the stations would be a paper of its own – and then we couldn't present new documentary data on droughts from the database. The referee does not like "precipitation index" in a paper of drought. We have used this term since most data considered have been about too little rain causing the drought. The referee asked for more details on the database, which we have tried to present in a better and more comprehensive way – where it comes from, how it was created and that it will soon be publicly available at the Bolin Centre for Climate Research. However, the database is in Swedish. We can't see how that could be changed, since the documents underlying the excerpts of the database are in Swedish. Should we translate the database? Translations are very tricky and misunderstandings and anachronisms would be bound to occur. Historical descriptions and data are of course bound within their historical, political, religious, legal and geographical context. Translation of a database of more than 20,000 excerpts is out of limit for the authors. The referee was very critical to our figures. Most of them have been taken away. The referee is not satisfied with our discussion and that we are too hesitant in describing the index as a drought index (and not as a "good-summer-weather-index"). We have reworked the paper and try to show that although correlation is higher between temperature and precipitation, the original data used ARE descriptions of drought or wetness rather than descriptions of hot or cold. But in Sweden, high temperatures results in dryness and cold temperatures in wetness. We are very grateful to both referees and they have much improved the paper. Yours truly, Lotta Leijonhufvud Dag Retsö

---

## Author Response (AR1)

**Editor Decision: Reconsider after major revisions** (20 Jul 2020) by Andrea Kiss
Comments to the Author:
Dear Authors,

thank you for the valuable manuscript. As the authors in their replies practically in all cases agreed, and willing to follow the clear and substantial comments and suggestions of the two referees, I only have a few, additional remarks: please, try to follow the requirements of the journal regarding the structuring of your paper (esp. regarding your "Results" chapter, but also there could be more and separate "Discussion"). Another remark is related to the referee comment on the uncertainties around the "database": this could be solved, for example, with a Supplementary table similar to e.g. the paper "Retsö 2015" in the HESS journal (about floods in Sweden roughly for the same period). There the lead author provided the key description of each event with a few words or max. a sentence. Lengthy text quotations, especially if it should be first translated from Swedish, does not necessarily provide more help to the readers than a short concise text. As for removing the figures: transforming/modifying figures to a more complex representation of the data and analysis discussed might be a better solution than to entirely remove those figures.
Thank you, and I look forward to read your revised manuscript.

**Reply to Editor Decision:**

Dear Editor,

We have now changed the following items:

1) A clearer distinction has been made between "Results" and "Discussion and conclusions".
2) A year-by-year supplementary table (similar to the one in Retsö 2015) has been provided and substitutes the previous period table.
3) The figures/graphs have been improved.

Best regards,

Dag Retsö and Lotta Leijonhufvud

---

## Referee Report (RR1)

Review of Leijonhufvud and Retsö: Documentary evidence of droughts in Sweden between the middle ages and c1800

The revised manuscript is improved compared to the first version. Clearly, the authors have made good efforts to meet several of the critical comments and to follow suggestions given by the two referees. Overall the paper reads better now and the new table at the end of the text provides valuable information summarized in a clear and simple way.

Nevertheless, there are still several problems that need to be addressed before the text is in a sufficiently good state for being accepted for publication. The problems can be divided into four main types: (1) there is still a lack of clarity in the description of how the drought index is constructed and also an associated lack of state-of-the-art discussion of the new index series in context of previous similar work in other regions, (2) there appears to be no connection between the detailed discussion of the evidence for droughts in the documentary data and the new drought index data; in particular the eight sub-periods that are concluded to be particularly struck by summer droughts, and even highlighted in the abstract, are not discussed in connection with the drought index data, (3) it is very difficult to follow and to understand the logic behind the way how the authors discuss their index series in comparision with instrumental data, and (4) the logical order of how some parts of the material is presented to the reader can be improved by some simple rearrangement and clarification of some of the text.

My overall recommendation is a major revision. However, the reason for labeling this as 'major' is mainly that there are so many individual items that need revision. I don't require any new analysis, but rather a quite large number of clarifications and some extended explanation and discussion. Most of the issues that need to be addressed are of rather minor nature. Thus, in terms of working efforts needed, the task of undertaking the revision satisfactorily may be regarded as major rather than minor.

Below I provide a detailed list in order of appearance in the manuscript, where each item is identified by its associated line number. Some items are clearly very minor. Some other items are of higher scientific importance. I don't start each sentence with a 'please', but that is what I mean. I encourage the authors to consider each item in this list. I am convinced that much can be done to help future readers to appreciate the material being presented in a better way. I certainly think this material deserves to be published, but not before another round of revision.

13. Explain here what kind of index that has been developed.

26. Before you can refer to "our drought index" for the first time in the text, you must tell the reader that you have created such an index and briefly explained what it is.

37-51. Insert at least one reference after each essential sentence in the introduction so that it becomes clear that the material presented there is based on knowledge from the scientific literature.

61. It is not acceptable to just say that database will be available somewhere later on. The database should be published in a trusted repository for research data before the manuscript is completed. Complete citation details including a digitical object identifier (DOI) should be provided so that future readers can find the database. Given that the data are from Sweden, a recommendable example of a trusted repository is the one provided by the Swedish National Data Service at https://snd.gu.se/en and https://snd.gu.se/sv

77. The term 'homogenized' only applies to the temperature data from Stockholm, but not for the precipitation data. However, the sentence on line 77-78 gives the reader the impression that all

instrumental data used have been homogenized. Please re-write the text to avoid confusion about this already at the start of this paragraph.

85. It is unclear what you mean when you say that the adjustment factors "will have no effect on correlation coefficients". The statement is possibly wrong. Correlation coefficients between two time series will indeed be affected to some extent if one of the two series is adjusted in a sub-period of the entire time period.

88. The headline 'Method' here is not well chosen. The text that follows until the next headline (which occurs on line 163) does not just describe a certain method, but rather first provides information about the drought index series being derived and then provides information about correlations between this index series and instrumental temperature series and also correlations between instrumental temperature and precipitation data. I suggest to choose a more descriptive headline, which only applies to how the drought index is derived. Such a headline could for example be:

Method for construction of a drought/precipitation index series

90-132. This is the part of the manuscript that describes how the drought index series is derived. It is clearly an imporant and central part of the text. Actually, the authors themselves even state on line 116 that "The most important part of the present analysis is the construction of an index." However, the current version of the text that describes how the index is derived has several problems. Given the importance of this section, more efforts should be made on improving the quality of the text and thereby also the clarity of how the index was constructed. In particular, the following should be addressed:

(a) Add a general discussion of the concept of climate indices as used in historical climatology in studies from other countries, with inclusion of references to key papers in order to give the reader a good view of the state-of-the-art. This should include at least some argumentation for why the current authors have chosen a 7-point index scale from -3 to +3 and not any other scale that in principle could have been chosen alternatively.

(b) The current text on lines 90-132 does not provide clear information that the index scale actually is a 7-point index scale from -3 to +3, or what is meant by each of the seven points on the scale. The only place in the entire manuscript where this is explicitly written is not found until in the very last few sentences in the final discussion (lines 423-426). I was very surprised to see this kind of information only at the end of the paper. It should of course be provided already in the "Method" section so that the reader knows what the index values mean when reading the rest of the paper. I strongly suggest that the authors insert a new table in the "Method" section, which clearly indicates what each of the seven points (-3, -2, -1, 0, +1, +2, +3) on the index scale means, and - this is very important - that provides information about the criteria that have been used in order to determine which point on the scale that is chosen for any given year. This is a main point of weakness of the manuscript in its current form; there is still no information about how the index data is created.

(c) Continuation of the previous point: The text on lines 90-132 does not make it clear whether the index values are based on information from the entire year (i.e. from any part of an individual year) or if they are based only on data from let's say summer, or spring-summer-autumn, or anything else. Please clarify this.

Figure 1. In my opinion, Figure 1, which shows the index data, should not be presented in the "Method" section, but rather in the "Results" section - because it is a result of the application of a method. Thus, I suggest to move Figure 1 and all text that is directly associated with this figure to

immedately below the headline "Results" that appears on line 332 in the current manuscript. Actually, I also suggest that the current Table 3 and everything that is now written on lines 420-429 (i.e. the last part of the final discussion), is also moved to the "Results" section. In my view, both Figure 1 and Table 3 illustrate results of the authors' work and they should thus be presented in the "Results" section.

134. I suppose the text on this line is meant to be a figure caption. If so, it should be moved to below the figure and also made sufficiently complete to be understandable without forcing the reader to read the entire method section.

141-160. The entire text in this part of the manuscript, which is about correlation between in particular the index data and instrumental temperature data, is in my view (just like Figure 1 and Table 3) a part of the result, and not a part of any method. So, I recommend to move this text and place it somewhere below the headline "Results", but after Figure 1. However, the text currently being placed on lined 141-160 has several problems. I simply have great difficulties in following the logic there so it should be re-written and clarified better. I think I do understand what the authors want to say, but the text is not constructed in a very helpful way. Here are examples of problematic issues:

141. State exactly which time period that is used for calculating correlations between instrumental temperature and precipitation data.

142. Why is the slightly negative correlation between JJA temperature and precipitation in Stockholm 'rather surprising'???

146. You could insert the word 'instrumental' before 'precipitation'.

146, and elsewhere. I recommend that you throghout the entire manuscript not write 'precipitation/drought' when you refer to instrumental data that are actually simply 'precipitation' data, but not some kind of combination of data for precipitation and drought.

148. The phrase 'turned out significantly' is not well chosen. If you mean that you find correlations that are statistically significant at a certain chosen level, then this level should be stated and the words should be better chosen to satisfy a reader who reads through a statistician's glasses.

151. The term 'non-existing' is also not well chosen. Do you perhaps mean that the correlation does not reach a certain significance threshold?? If so, which threshold?

153-154. The sentence on these lines is identical to the first part of the sentence on lines 147-149. Delete the sentence on lines 153-154.

156-158. I find the explanation to Table 1 not sufficently clear. Here is my guess of what you mean:

Correlation coefficients calculated between average monthly and seasonal temperatures in Stockholm and (on the first row) with the drought index 1756-1816 and (on the second row) with the corresponding monthly and seasonal precipitation data from Stockholm for the period 1859-2011.

I also suggest that you move and slightly re-phrase the text in the parenthesis (which is better written: daily observations are summed to monthly or seasonal values) to be placed below the table, immediately after your full reference to the precipitation data. Or, this information may instead be given only at the end of the paper, where you provide references to your data sources.

Table 1: The last column labled 'C-Scan' is not explained anywhere. It should be deleted (or be explained).

163-330. This part of the manuscript contains detailed information about what kind of information about droughts (and related things) that is found in the sources. However, it is still entirely unclear, at least to me, how this information has been used to define the index values. Thus, I don't see any clear connection between all the detailed information on lines 163-330 and the method description (currently on lines 90-132). Please, do something that helps me and other readers to understand better how you have derived your index values. This is actually a very important aspect. As I already pointed out above, the authors state on line 116 that "The most important part of the present analysis is the construction of an index." Thus, it is very suprising that I, as a reader, cannot find any connection at all between the detailed description of the content about droughts in the docuementary evdence provided on lines 163-330 and the derived index data. Therefore, I must say that I get the impression that one of the two authors wrote the text on lines 163-330 and the other author wrote the rest of the paper and that there is no connection between these parts. In particular, I find it astonishing that the eight sub-periods of drought in the 17th and 18th centuries are not at all associated with any discussion of their corresponding values on the index scale. Were these eight periods identified entirely without using the index data??? If this was so - then what is the scientific value of the index data in this study??? If I look in Figure 1, I can see that all the eight mentioned drought periods coincide well with time periods when the gaussian filtered data lie above +1 on the index scale. Is this perhaps something the authors have considered? In any case, the lack of clear connection between the detailed discussion on lines 163-330 and the construction of the index, as well as how the eight drought periods were defined, is another main point of criticism to the manuscript in its current form. So, please, help the reader and put the pieces better together.

166. It could be helpful to insert a sentence like the following one, immediately after the sentence on lines 165-166:

Therefore, no attempt has been made here to derive any drought index values before 1500.

271. How did you identify the eight periods? Which criteria were used? It is important to provide clear information about this, given that you have chosen to structure so much of your text on this, inluding the abstract and the content of Table 4.

333. As I suggested above, both Figure 1 and Table 3, with their associated texts, as well as the text that is currently placed on lines 141-161 is about your results and would better be placed here.

334. For clarity, insert 'in Stockholm' after 'summer temperatures'. Also insert 'for Sweden' after 'the drought index'.

334-335. The entire sentence "The correlation from Table 1 of 0.47 is expressed as R² in Figure 2" is entirely non-understandable. Either delete it, or re-write in an understandable way.

335-337. Because the sentence on lines 334-335 is non-understandable, also the text on lines 335-337 is non-understandable. Either delete it, or re-write in an understandable way.

339. I suppose this is meant to be a figure caption and should so be placed below the figure. Also, for clarity, insert 'Stockholm' before 'JJA temperature'.

346-348. This sentence is difficult to understand. The logic is unclear. Please explain better what you mean.

348. Because of the difficulty to understand what is written on lines 346-348, it is impossible to understand what you mean with "We believe the main reason for this...". Please explain better what you mean.

349. What do you mean with 'more stringent' here?

354-355. The text here implies that you have computed correlation coefficients between instrumental precipitation data and the index data, but you have not presented any results that show this. So, this text is quite non-understandable here. Please explain better what you mean.

361. I think the text reads better if you replace "are compared to average monthly temperature for the entire period" with "are compared to average corresponding monthly or seasonal temperatures for the entire period"

362. In what sense does none of the dry sub-periods differ "significantly" from the average monthly temperatures...? If you don't mean "significantly" in any statistical sense, then perhaps the word "notably" would suit better. But even so, I personally think a difference by more than 1 degree Celsius between one sub period and the whole period is quite a substantial difference. Please be more careful with how you use the term "significantly", which can be interpreted in different ways depending on the reader's profession.

Table 2. I don't understand what the values given within parentheses on every second row mean or how theywere obtained. This should be explained clearly. If these values represent averages of the index values, then I don't understand how you can get different numbers in June, July, Aug and JJA - because I have got the impression that your index series always has exactly one value in each year, and thus it is impossible to obtain different values for different months within a given sequence of years. Even more confusing: If the values within parentheses are some kind of mean values for the drought index, then how can the value for JJA during 1757-1767 be below 1.00, when the corresponding values for the individual months are above 1.00. Something needs to be clarified and/or corrected here.

379-380. The way you write the first sentence in the discussion implies that you have tried to calculate correlation coefficients between your drought index series and instrumental drought data (and to instrumental temperature data). However, this gives an incorrect impression because you have not computed any correlations between your index data and any drought data. Please, re-write and explain better what you mean and what you have done.

385. "The main problem" with your precipitation/drought index series is certainly not that "you" only have a short period with overlapping instrumental data of precipitation/drought. But it is true that it is difficult to try and translate your index into let's say seasonal precipitation sums. Please, re-write and explain better what you mean. I would also welcome a more extended discussion of what really is "the main problem" of your drought index - such a discussion could preferably refer to problems and difficulties already discussed in other similar studies.

387-388. Here, again, your text implies that you have calculated correlation coefficients between your index series and instrumental precipitation data - but clearly you have not done that. Or, maybe, you have actually done such caluclations - but you have not reported them in your result section?? Please, re-write and explain better what you mean.

405. What exactly do you mean when you say that "Stockholm temperatures after 1756 have showed to be positively biased"?

406. What is "TRW" and what is "density"?? And what has this to do with your drought index data???

415-429. As explained above, I recommend that this bit is moved to the "Results" section.

431. It would be good to end your discussion by trying to provide an answer to your main question posed in the introduction on lines 26-27, i.e.: "Is it possible to distinguish periods of drought in Sweden through documentary sources from the 15 th till the 18 th century?".

449. I suggest to insert the word 'instrumental' between 'The' and 'datasets'.

691. This table should be labeled Table 4 (not Table 3). The table is certainly very informative. I think it would be excellent if you could extend the table and provide the same kind of information also for all other years in the entire period 1500-1816. That would really make it possible to see and understand how you have derived every single index value in your data series! (But I do understand if you think this will take too much time and my suggestion here is just a suggestion and not a requirement.)

Finally, in the spirit of FAIR data and Open data, the entire index series from 1500-1816 should be provided in the form of data that can be used by others. The simplest way is probably to just present them in a single table in the paper. A better way, however, would be to publish your index data in a trusted repository for research data, so that the data are easily findable and citable. Given that the data are from Sweden, a recommendable example of such a repository is the one provided by the Swedish National Data Service at https://snd.gu.se/en and https://snd.gu.se/sv. Thus, both the database and the index series could be published as separate datasets in (for example) the SND catalogue, each with its own unique address and unique DOI.

---

## Author Response (AR2)

We have now updated our latest version, to the greatest degree as possible according to the last reviewers comments.